# Nebulized Lipopolysaccharide Causes Delayed Cortical Neuroinflammation in a Murine Model of Acute Lung Injury

**DOI:** 10.3390/ijms251810117

**Published:** 2024-09-20

**Authors:** Katharina Ritter, René Rissel, Miriam Renz, Alexander Ziebart, Michael K. E. Schäfer, Jens Kamuf

**Affiliations:** 1Department of Anesthesiology, University Medical Center of the Johannes-Gutenberg-University, Langenbeckstrasse 1, 55131 Mainz, Germany; 2Research Center for Immunotherapy (FZI), Johannes-Gutenberg-University, 55131 Mainz, Germany; 3Focus Program Translational Neurosciences (FTN), Johannes-Gutenberg-University, 55131 Mainz, Germany

**Keywords:** lung injury, lipopolysaccharide, neuroinflammation, mice

## Abstract

Lung injury caused by respiratory infection is a major cause of hospitalization and mortality and a leading origin of sepsis. Sepsis-associated encephalopathy and delirium are frequent complications in patients with severe lung injury, yet the pathogenetic mechanisms remain unclear. Here, 70 female C57BL/6 mice were subjected to a single full-body-exposure with nebulized lipopolysaccharide (LPS). Neuromotor impairment was assessed repeatedly and brain, blood, and lung samples were analyzed at survival points of 24 h, 48 h, 72 h, and 96 h after exposure. qRT-PCR revealed increased mRNA-expression of *TNFα* and *IL-1β* 24 h and 48 h after LPS-exposure in the lung, concomitantly with increased amounts of proteins in bronchoalveolar lavage and interstitial lung edema. In the cerebral cortex, at 72 h and/or 96 h after LPS exposure, the inflammation- and activity-associated markers *TLR4*, *GFAP*, *Gadd45b*, *c-Fos*, and *Arc* were increased. Therefore, single exposure to nebulized LPS not only triggers an early inflammatory reaction in the lung but also induces a delayed neuroinflammatory response. The identified mechanisms provide new insights into the pathogenesis of sepsis-associated encephalopathy and might serve as targets for future therapeutic approaches.

## 1. Introduction

Community-acquired pneumonia is a major cause of hospitalization and mortality worldwide while hospitality-acquired pneumonia remains one of the most dominant nosocomial infections [1,2,3,4]. Infectious diseases of the respiratory system are a leading origin of acute lung injury (ALI) and sepsis, with a mortality of more than 50% in pneumonia-related septic shock [5,6]. Sepsis-associated encephalopathy (SAE), ranging from mild delirium to deep coma, occurs in up to 70% of patients suffering from severe systemic infection and is associated with increased morbidity and mortality [7,8,9]. However, the pathogenesis of SAE and sepsis-associated delirium (SAD) is poorly understood. Ischemia, alterations in cerebral metabolism and microcirculation [10], impaired neurotransmission, dysfunctional synaptic plasticity, and a compromised blood–brain barrier [11] which is subjected to increased levels of pro-inflammatory cytokines constitute only a small fraction of the underlying mechanisms [12,13]. Lipopolysaccharides (LPSs) are a major component of the outer membrane of Gram-negative bacteria and stimulate the inflammatory response as pathogen-associated molecular patterns (PAMPs), mainly via Toll-like receptor 4 (TLR4)-mediated signaling [14]. Applications of LPS via injection or intratracheal instillation are common methods to induce acute lung injury (ALI) and/or sepsis in experimental settings and have been proven to induce an intense local and systemic inflammatory response and pulmonary tissue damage [15,16,17,18,19]. Physiological dysfunction, alveolar capillary leak, infiltration of immunologically active cells, and pro-inflammatory cytokine expression mimic the pathogenesis of ALI in bacterial pulmonary infection and sepsis of other origin [20,21]. The brain—apart from intracerebroventricular (i.c.v.) application of LPS—rarely serves as the primary site of stimulation, but it is still affected as neuroinflammation is frequently observed after intraperitoneal (i.p.) or intravenous (i.v.) injection [22,23]. Cognitive impairment associated with hippocampal cell loss and microglial activation, along with increased levels of interleukin-1β (IL-1β), tumor necrosis factor α (TNFα), and nitric oxide (NO), was observed in murine brains after i.p. and i.c.v. administration of LPS, while interleukin-4 (IL-4) and interleukin-10 (IL-10) were reduced [23,24,25]. Furthermore, chronic administration of LPS serves to induce experimental Alzheimer’s disease (AD) and depression, and LPS was detected in amyloid plaques of AD, which connects also it to long-term neurocognitive disorders [26,27].

However, the majority of existing data on the LPS-induced neuroinflammatory response are based on the systemic application of the bacterial endotoxin. In the present study, we tested the hypothesis that an inflammatory stimulus primarily directed to an organ separated from the brain by the alveolar–capillary and blood-brain barriers can induce neuroinflammation. To address this issue, mice were subjected to a single full-body-exposure to nebulized LPS and organ-specific responses in the lung, blood, and brain were investigated over a period of 96 h.

## 2. Results

### 2.1. Single Exposure with Nebulized LPS Leads to Prompt Induction of Inflammatory Response in the Lung

Interleukin-6 (*IL-6*) and tumor necrosis factor alpha (*TNFα*) mRNA expression in lung tissue samples analyzed by qPCR and was significantly increased at 24 h (*IL-6* 0.0022 ± 0.0004, *TNFα* 0.0007 ± 9.678 × 10^−5^) and 48 h (*IL-6* 0.0011 ± 0.0002, *TNFα* 0.0002 ± 5.42 × 10^−5^) after LPS nebulization in comparison to the vehicle (*IL-6* 9.81 × 10^−5^ ± 8.32 × 10^−6^; *TNFα* 3.29 × 10^−5^ ± 3.24 × 10^−6^; *p* (24 h) < 0.0001, *p* (48 h) = 0.028) and compared to 72 h (*IL-6* 0.0002 ± 2.32 × 10^−5^, *TNFα* 4.37 × 10^−5^ ± 4.60 × 10^−6^; *p* (24 h) < 0.0001, *p* (48 h) = 0.028) and 96 h after exposure (*IL-6* 0.0001 ± 3.11 × 10^−5^, *TNFα* 3.35 × 10^−5^ ± 3.60 × 10^−6^; *p* (24 h) < 0.0001, *p* (48 h) = 0.026) (Figure 1A,B).

Histological scoring performed for edema, epithelial destruction, microatelectasis, and overdistension (Figure 1D) revealed significantly increased interstitial edema at 24 h (3.13 ± 0.21 pts.), with fading intensity from 48 h (2.53 ± 0.17 pts.) after LPS exposure in comparison to the vehicle (2.20 ± 0.25; *p* (24 h) = 0.033). The tissue-free area was reduced at 24 h (61.23 ± 1.91%) in comparison to the veh (66.61 ± 1.38%; *p* = 0.047), and at 72 h (67.25 ± 1.34%; *p* = 0.032) and 96 h (70.40 ± 0.90%; *p* = 0.0005) after nebulization (Figure 1C). The protein amounts in BAL samples, quantified by the Bradford assay, were significantly elevated 24 h (0.863 ± 0.129 mg/mL) and 48 h (0.625 ± 0.099 mg/mL) after LPS exposure compared to the vehicle (0.283 ± 0.057 mg/mL; *p* (24 h) = 0.0006, *p* (48 h) = 0.046), 72 h (0.381 ± 0.058; *p* (24 h) = 0.003), and 96 h (0.336 ± 0.026; *p* (24 h) = 0.002) (Figure 2E). Taken together, these results show a strong immediate inflammatory peak caused by the single injection to nebulized LPS, subsequently decreasing to baseline three days after exposure.

### 2.2. Single Exposure with Nebulized LPS Is Sufficient to Generate a Systemic Bloodstream Translocation

Inhalation of LPS through the physiological airway led to pulmonary inflammation and induced pro-inflammatory and potentially neurodetrimental gene expression in the cerebral cortex. The concentration of LPS-binding protein (LBP) in blood samples was analyzed using ELISA to determine its translocation from the lung to the bloodstream. The plasma levels of LBP were significantly higher 24 h (935.3 ± 58.9 ng/mL) and 48 h (666.4 ± 60.54 ng/mL) after nebulization compared to 72 h (445.5 ± 12.73 ng/mL; *p* (24 h) < 0.0001, *p* (48 h) = 0.0033), 96 h (489.1 ± 26.06 ng/mL; *p* (24 h) < 0.0001, *p* (48 h) = 0.0231) and vehicle (524.4 ± 23.36 ng/mL; *p* (24 h) < 0.0001, *p* (48 h) = 0.07), indicating circulating LPS as a potential link between lung and brain (Figure 2D).

### 2.3. Single Exposure with Nebulized LPS Leads to Delayed Neuroinflammatory Response in the Cerebral Cortex and Hippocampus

Neurological state was evaluated using NSS at 24 h intervals starting 24 h prior to LPS exposure (Figure 2B). Mice exposed to LPS displayed an increased NSS on the same day of nebulization (1.33 ± 0.12 pts.) compared to vehicle group (0.20 ± 0.13 pts., *p* = 0.015) and decreased performance compared to their pre-exposure status (0.55 ± 0.11 pts., *p* = 0.0005), which returned to normal levels 24 h post-LPS administration. Although slight body weight loss was observed in all LPS-exposed mice after 24 h, this finding was not statistically significant (Figure 2C).

qRT-PCR was performed for several (neuro-)inflammatory targets in brain tissue samples of cortex (Figure 3A) and hippocampus (Figure 3B) from all analyzed survival points. *IL-6* was chosen as acute phase marker. Toll-like-receptor 4 (*TLR4*) is the main receptor for LPS, which mediates pro-inflammatory signaling in various neurological disorders [28] and glial fibrillary acidic protein (*GFAP*) is a common marker for astrocyte activation [29]. Activity-regulated cytoskeletal gene (*Arc*), *c-Fos* and growth arrest and DNA damage inducible beta (*Gadd45b*) are immediate early genes (IEGs) and are considered as markers upregulated in response to physiological and environmental stressors. While *Arc* is predominantly expressed by neurons and critically involved in synaptic plasticity [30], *c-Fos* and *Gadd45b* are expressed by neurons and glia, serving both as markers for neuronal and glial activation [31,32]. While mRNA-expression of *IL-6* was unaffected by the exposure to nebulized LPS, expression of *GFAP*, *Arc*, *c-FOS* and *Gadd45b* was significantly increased 72 h and 96 h in cortical samples after LPS in comparison to the vehicle group and to early post-exposure time points of 24 h and 48 h, indicating a delayed astroglial and neuroinflammatory activation in the cerebral cortex following the early pulmonary response. qPCR analysis in the hippocampal tissues did not yield comparable findings, although increases in the mRNA expression of *Gadd45b* and *Arc* were observed. Detailed results are listed in Appendix A.

To analyze astrocytic response in this specific region of interest, immunohistological staining for GFAP was performed (*n =* 5 of each group). Mice showed an increased number of immunopositive particles in the dentate gyrus from 48 h after exposure, reaching a trend towards statistical significance (*p* = 0.065) compared to the vehicle group (0.57 ± 094) 96 h (1.76 ± 0.16) after LPS exposure (Figure 4).

## 3. Discussion

The present study aimed to investigate whether an LPS-mediated inflammatory stimulus on the lung is capable of inducing neuroinflammation in the brain. The main LPS mechanisms of stimulating immune pathways, including its binding to TLR4 by interacting with immunomodulatory factors LBP, CD14 and myeloid differentiation protein 2 (MD2) appear well understood [14]. Intercellular adaptor proteins like MyD88 activate kinase regulated signaling cascades, which stimulate NFκB pathway and therefore induce the expression of pro-inflammatory cytokines like TNFα [33]. In the present study, we detected a strong increase in the mRNA-expression of TNFα and IL-1β in lung tissue 24 h after exposure to nebulized LPS, which diminished 48 h and reached expression levels of the vehicle group 72 h after exposure. These findings are consistent with previous reports, characterizing the dynamic profile of inflammatory cytokine expression after LPS as a prompt and pronounced peak reaction to the stimulus with a rapid decrease to pre-exposure status [34]. In the early phase (24 h), we also observed quantifiable histological damage in form of interstitial edema, a common pathology linked to LPS-induced lung injury [35,36]. A major limitation of the study is that the amount of LPS taken up by each animal remains unknown. The concentration of the aerosol is estimated, yet the amount of circulating LPS that depletes in the lung is not determined. Considering the calculated concentrations and duration of exposure, the LPS dose in this study might potentially exceed that of more invasive models utilizing intratracheal instillation or i.p. injection [37,38]. However, none of the animals in this study met termination criteria and we observed a significant decrease in the inflammatory response in the lung after 48 h. Taken together, the single full-body-exposure to nebulized LPS provides a sufficient inflammatory lung injury and therefore represents a less invasive alternate model to the predominantly used intratracheal instillation. Future studies are needed to analyze dose dependent effects on the inflammatory response and to quantify the exact amount of LPS supply.

Neuromotor impairment was analyzed by NSS, a tool frequently used in models of experimental TBI [39,40,41]. All mice subjected to nebulized LPS showed significantly increased scores compared to vehicle at the day of exposure, yet this effect faded rapidly within 24 h. While NSS serves well in discriminating the degree of motoric dysfunction after a localized cerebral injury, its capacity of capturing higher cognitive functions is strictly limited. Most studies about LPS-induced neuroinflammation or SAE include assessment of spatial learning, long-term memory and emotional learning [22,23,27]. As the observed impact on the NSS appeared at the day of exposure and far before the upregulation of neuroinflammatory gene expression, it is most likely confounded by a temporary debilitated general condition. Therefore, not addressing more elaborated functions in neurobehavioral testing is considered as a major limitation of this study.

Our analyses revealed increased expression of *GFAP*, *Arc*, *Gadd45b* and *c-Fos* in the cerebral cortex at 72 h and 96 h after exposure. The increased expression of *GFAP* indicates the activation of astrocytes [29], which was induced by i.p. administration of LPS and connected to NFκB downstream signaling in previous works, while in vitro settings revealed CD14 as a dominant co-factor in TRL4 mediated astrocytic stimulation [42,43,44]. However, the delayed up-regulation of *GFAP* mRNA-expression was only observed in the cortex, but not in the hippocampus. A similar regulation was detected for the mRNA-expression of the IEGs *c-Fos* and *Gadd45b*, which serve as indicators for cellular responses to environmental stressors [31,32]. *c-Fos* was shown to be upregulated after LPS stimulation in primary glia cultures and astrocytes in vitro [45,46], whereas upregulation of *Gadd45b* was associated with anti-apoptotic processes in astrocytes [47]. Together, the mRNA-expression profile of *GFAP*, *c-Fos* and *Gadd45b* indicate a delayed glial activation in the cerebral cortex in response to nebulized LPS exposure.

*Arc*, which is also classified as an IEG, has been characterized as a reliable indicator of neuronal activity and synaptic plasticity is deeply involved in glutamatergic and dopaminergic signaling, memory acquisition and cognitive flexibility [30,48,49,50]. We found that *Arc* mRNA-expression was upregulated in the cerebral cortex at 72 h and 96 h and in the hippocampus at 72 h after LPS exposure. Other data regarding the short-term regulation of *Arc* in the murine brain following LPS exposure are scarce and the effects on neural circuit function remain unknown. However, chronic intracerebroventricular infusion of LPS for 28 d in rats resulted in increased numbers of *Arc*^+^ neurons in hippocampal subregions, which further increased following the completion of explorative tasks [50,51]. Conversely, the downregulation of *Arc* was reported one month following intraperitoneal injection of LPS and associated with depressive- and anxiety-related behavior [52]. Together with our results, these findings indicate a dynamic regulation of *Arc* mRNA-expression in the brain response to LPS exposure. Further investigations are necessary to elucidate the relationship between spatiotemporal expression regulation and functional implications of *Arc* and the other gene expression markers used in this study. Taken together, we detected a delayed upregulation of several markers of neuronal and glial activation and inflammation in the cerebral cortex, but also in the hippocampal area following the induction of a pulmonary inflammation by nebulized LPS.

As mentioned above, the majority of experimental works investigating SAE refer to i.p. administration of LPS and describe a rapid increase in inflammatory markers in the brain. Here, we demonstrate that targeting the lung as a primary focus is capable of inducing neuroinflammation, yet the cerebral response occurs with a delay of not less than 72 h. This study worked with a consistent dose of LPS and therefore cannot provide a statement about the onset of neuroinflammation in higher or lower doses of LPS. Systemically applied LPS evokes a wide spectrum of particularly detrimental cellular pathways in the BBB. Destruction of tight junction proteins, increase in caspase-mediated apoptosis and upregulation of intercellular adhesion-molecules provide a gateway for the circulating LPS [53]. In critical review, the way of administration chosen in this study bears the potential of transferring LPS directly to the brain via the nasal neuroepithelium. Intranasal administration is a successful and less invasive route of drug administration frequently used in neuroscientific experiments [54,55]. However, administration of LPS i.p. or i.c.v. induces a rapid expression of-pro-inflammatory cytokines and TLR4 in the brain up to 24 h or 48 h after administration [56,57,58]. In the present study, mRNA-expression of IL-6 was entirely unaffected by the LPS stimulus in both analyzed cerebral regions, while the increase in *TLR4* expression was detected 72 h after exposure, settled right between the early inflammatory peak in the lung and the delayed neuroinflammatory response. The expression dynamics alongside the increased plasma levels of LBP—which exerts an essential function in facilitating the binding of LPS to its receptor [33]—in the present study undermine the theory of a translocation of the inflammatory stimulus from the lung via the circulating blood stream to the brain [22].

## 4. Materials and Methods

### 4.1. Animals and Study Groups

All experiments were conducted under approval of the responsible animal welfare committee of the Landesuntersuchungsamt Rheinland-Pfalz (23177-07/G17-1-088) and in accordance with current national and international guidelines [59]. 70 female C57BL/6 mice (Jackson Laboratory, Bar Harbor, ME, USA), 10 weeks old with an average body weight of 17–22 g were housed in groups of four or five under standard conditions (12 h light/dark cycle, 22–24 °C, 55% humidity) with access to food and water ad libitum. Mice subjected to nebulized lipopolysaccharide from *Escherichia coli* (Sigma Aldrich, St. Louis, MO, USA) 15 mg/5 mL NaCl 0.9% or vehicle solution (NaCl 0.9%; veh, *n =* 10). Nebulization was performed by a medical nebulizer (IH50, Beurer, Ulm, Germany) placed in the temporarily sealed cage with a volume of 8.5 L for 30 min as full-body-exposure. Process of nebulization required 15 min, leading to concentrations from 0.11 μg LPS/mL air at the beginning to a maximum of 1.76 μg LPS/mL air at full nebulization, and mice rested in the aerosol for 15 min hereafter. Considering the standard respiratory rate of adult mice this leads to a theoretical maximum of approximately 52 μg LPS/min circulating through the airway [60]. Mice were randomly assigned to a survival period of 24 h, 48 h, 72 h or 96 h (each *n =* 15). Neuromotor deficit was assessed daily by Neurological Severity Score (NSS, 0–15 points), including assessment of coordination and balancing skills, general behavior, and motoric impairment by an investigator blinded to the protocol, and mice were screened repetitively for termination criteria, predefined as ≥20% loss of initial body weight and/or exhibiting signs of severe pain or discomfort [61]. Euthanasia was performed at the end of each specific survival period by cervical dislocation under isoflurane anesthesia (4 Vol% for 90 s), and tissue samples were gathered for further analyses.

### 4.2. Histological Analyses

After euthanasia, lungs were removed, fixed in paraformaldehyde (PFA) 4%, and embedded in paraffin (24 h, 48 h, 72 h each *n =* 6, 96 h, *n =* 5). Samples were cut into 4 μm sections and stained with hematoxylin–eosin according to the manufacturers’ protocol (Merck, Darmstadt, Germany). Sections were then analyzed by an investigator blinded to the treatment condition and survival time, following a predefined histological scoring system of five categories, including alveolar edema, interstitial edema, epithelial destruction, microatelectasis, and overdistension, with each category reaching from 0 (=none) to 5 (=maximum manifestation) points by using the Axiovert 200 microscope (Zeiss, Oberkochen, Germany, 20× objective). Additionally, sections were analyzed by using ImageJ Version 1.52a and the tissue free area was quantified as in indirect marker for the gas exchange surface using the image settings “edit: invert” and “adjust threshold: Huang”.

Brains were removed in toto, frozen in powdered dry ice, and stored at −20 °C. Brains were cut to 12 μm coronal slices at 16 consecutive brain levels with a distance of 500 μm, starting at bregma +3.14 mm using a cryotome (Cryo-Star NX70, Thermo Fisher Scientific, Waltham, MA, USA) and collected on Superfrost^®^ Plus Slides (Thermo Fisher Scientific Inc., Waltham, MA, USA).

3,3′-Diaminobenzidine (DAB)-based immunostaining was used to detect GFAP-immunopositive particles in the right hippocampus, as previously described [62,63]. Cryosections were fixed in 4% paraformaldehyde in phosphate-buffered saline (PBS), rinsed in PBS, and incubated in 3.0% H_2_O_2_. Sections were incubated with blocking solution (5% normal goat serum, 1% bovine serum, and 0.1% Triton-X100 in PBS) for 1 h at room temperature (RT). Primary antibody (rabbit anti-GFAP Z033401-Z, DAKO, Santa Clara, CA, USA, 1:500) was applied to the blocking solution and incubated at 4 °C overnight. The next day, sections were washed in PBS and incubated with secondary antibody (goat anti-rabbit (H + L) biotinylated #BA-1000-1.5, 1:50, Vector Laboratories, Newark, CA, USA) for 1.5 h at RT. Samples were then incubated with avidin–biotin complex (1 drop avidin, 1 drop biotin, 2.5 mL PBST; Vectastain ABC Kit, Vector Laboratories) for 2 h at RT, washed, and incubated with 3,3′-diaminobenzidine (1 drop for 1 mL PBS). Sections were washed and air dried, fixed in Xylol (Pancreac AppliChem, Darmstadt, Germany), and mounted in Entellan (VWR, Darmstadt, Germany).

Images of the cortex and right hippocampus were captured from two brain sections (bregma −1.86 mm to −2.86 mm) of each animal using a laser scanning microscope (Axiovert 200, 10× objective, Zeiss, Oberkochen, Germany). ROI of individual images was 0.52 × 0.65 mm, and anti-GFAP-immunofluorescence was quantitatively assessed by optical density measurement and cell counting using ImageJ Version 1.52a and the “Analyse Particle” plugin [62].

### 4.3. Gene Expression Analyses

Tissue samples of the lung were collected after euthanasia, and cortical and hippocampal brain regions samples were taken from the coronal sections of all mice (bregma +0.64 mm to −2.86 mm) during histological processing, snap frozen in liquid nitrogen, and stored at −80 °C. RNA extraction and cDNA synthesis were performed by using an RNeasy Kit and QuantiTect Reverse Transcription Kits (both Qiagen, Hilden, Germany), according to the manufacturer’s instructions. RNA extraction and cDNA synthesis were performed using an RNeasy Kit and QuantiTect Reverse Transcription Kits (both Qiagen), according to the manufacturer’s instruction. Gene target sequences were amplified and examined by qRT-PCR (LightCycler 481, Roche Molecular Systems Inc., Pleasanton, CA, USA) using Absolute Blue qPCR SYBR Green Mix Plus ROX Vial (Thermo Fisher Scientific) or LightCycler^®^ 480 Probes Master (Roche Molecular Systems Inc., Pleasanton, CA, USA). Samples were analyzed in duplicate, and quantification was performed using a target-specific standard curve and normalization to the reference gene cyclophilin A (Ppia). The sequences of applied primer pairs (5′–3′) (Table 1) are as follows.

### 4.4. Protein Assays

Samples of whole blood were taken after euthanasia and decapitation, supplemented with 80 μL of heparin sodium (5000 I.U./mL, Ratiopharm, Ulm, Germany), and centrifuged for 8 min at 3800 rpm; then, the extracted plasma was stored at −80 °C. Samples were diluted 1:200, and the concentration of lipopolysaccharide-binding protein (LBP) was analyzed using the LBP Immunoassay (LBP Mouse ELISA-kit KH 205-01, HycultBiotech, Uden, The Netherlands), according to the manufacturer’s instructions. Samples were analyzed in duplicate, absorbance was determined at 450 nm using a microplate reader (MRX TC II Microplate Absorbance Reader, Dynex Technologies, Chantilly, VA, USA), and the concentration of LBP was determined and expressed as ng/mL.

After euthanasia, surgical access to the trachea was carefully prepared caudal to the decapitation line, and a 20 G peripheral venous catheter (Jelco, Smith Medical Inc., Minneapolis, MN, USA) was inserted. Then, 1 mL PBS was slowly flushed in the lungs, aspirated, and stored at −80 °C paraffin (24 h, 48 h, 72 h each *n =* 6, 72 h, *n =* 7), and the protein concentration in bronchoalveolar lavage was determined by Lowry assay. Samples were homogenized in RIPA lysis buffer (50 mM Tris-HCl, pH 7.5, 150 mM NaCl, 1 mM EDTA, 1% (*v/v*) NP-40, 0.1% (*v/v*) sodium dodecyl sulfate, complete protease inhibitors (Roche)), incubated for 30 min, and centrifuged for 20 min. Then, 5 μL of the supernatants were supplemented with reagents A (#5000113), B (#5000114), and S (#5000115, all BioRad, Hercules, CA, USA), and protein concentration was determined by using a GloMax^®^-Multi Detection system (Promega, Walldorf, Germany).

### 4.5. Statistical Analyses

As this is an exploratory study, previous data, which could be used to calculate group sizes, are rare. The required sample size (*n =* 15 animals in LPS groups, *n =* 10 in the vehicle group) was determined based on the assumption that a change of 20% in the analyzed criteria would be relevant. The probability of type I error was set to a = 0.05, and the standard statistical power was set to 1 − b = 0.8 (80%), resulting in b = 0.2 (probability of type II error). A larger amount of animals was used in the LPS groups due to the natural range of inflammatory response, which was not to be expected in the vehicle group.

All data were analyzed using GraphPad Prism software (version 9, GraphPad Software Inc., San Diego, CA, USA). Outliers were identified using Grubb’s test and excluded from further evaluation, and the data distribution was analyzed by the Shapiro–Wilk normality test and QQ plots. Comparative analysis of more than two groups was performed by ordinary one-way analysis of variance (ANOVA) or Kruskal–Wallis test, followed by Holm–Sidak’s or Dunn’s multiple comparison test depending on the data distribution. Parameters more often evaluated at multiple time points in specific behavioral testing and body weight measurements were calculated using two-way ANOVA followed by Holm–Sidak’s or Tukey’s multiple comparison test. Values are presented as mean ± standard error of the mean (SEM); * *p* < 0.05, ** *p* < 0.01, and *** *p* < 0.001.

## 5. Conclusions

Taken together, this study confirms a single full-body exposure to nebulized LPS as an effective inflammatory stimulus to the lung, able to cross the barriers between lung, brain, and circulating bloodstream. The specificity of this study lies in the method of application of the pro-inflammatory stimulus and the time lapse between the inflammatory responses of the particular organs, as neuroinflammation-associated gene expression in the brain started to appear 72 h after the exposure. These findings might enlighten the pathologic background of neurocognitive disorders in acute lung injury. Further work is required to explore the impact on neurocognitive function and identify potential therapeutic approaches.

## Figures and Tables

**Figure 1 ijms-25-10117-f001:**
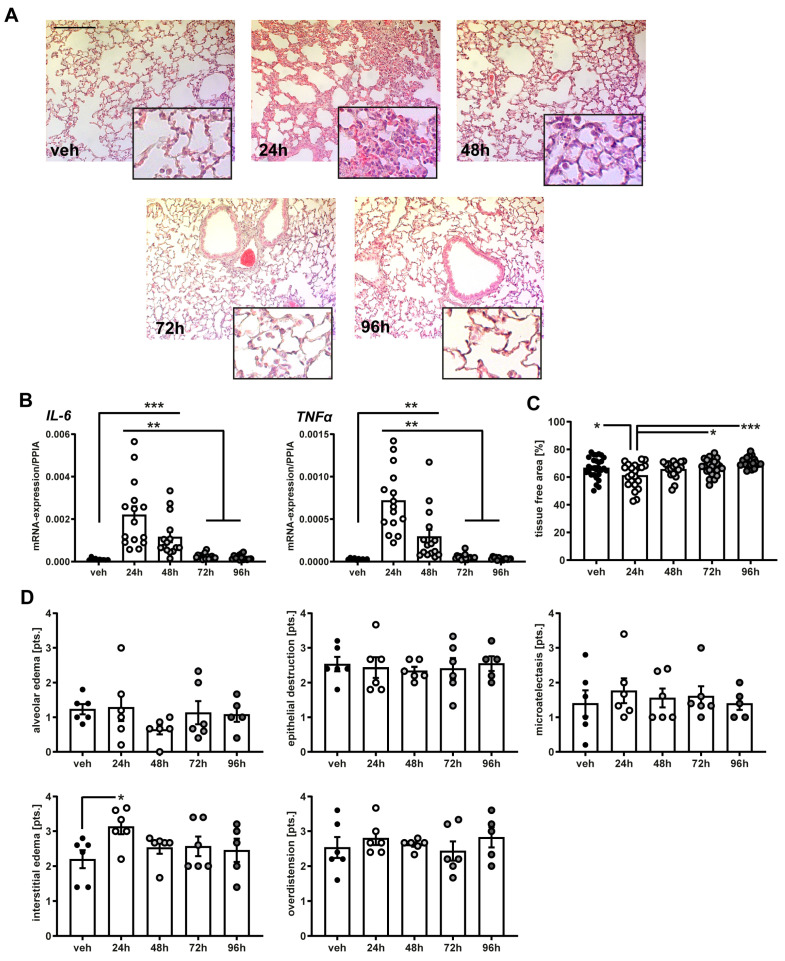
Single exposure to nebulized LPS induced inflammatory gene expression and histological damage in the lung. (**A**) Representative images of lung tissue samples in HE staining of all study groups. Scale bar: 500 μm. (**B**) Gene expression analyses were performed in the lung tissue samples of all study groups and normalized to PPIA. mRNA-expression of *IL-6* and *TNFα* were significantly increased 24 h and 48 h after LPS exposure in comparison to the vehicle as well as 72 h and 96 h after nebulization. (**C**,**D**) Histopathological damage was assessed by measurement of the tissue-free area in HE-stained lung sections and a semiquantitative scoring system in different section levels of each specimen (veh, 24 h, 48 h, 72 h each *n =* 6, 96 h, *n =* 5). Interstitial edema was significantly increased 24 h after LPS exposure compared to the vehicle, and tissue-free area was reduced at 24 h in comparison to veh, 72 h, and 96 h after nebulization. * *p* < 0.05, ** *p* < 0.01, *** *p* < 0.001. Values of all data represent mean ± SEM; *p* values were calculated by one-way-ANOVA followed by Holm–Sidak’s multiple comparison test.

**Figure 2 ijms-25-10117-f002:**
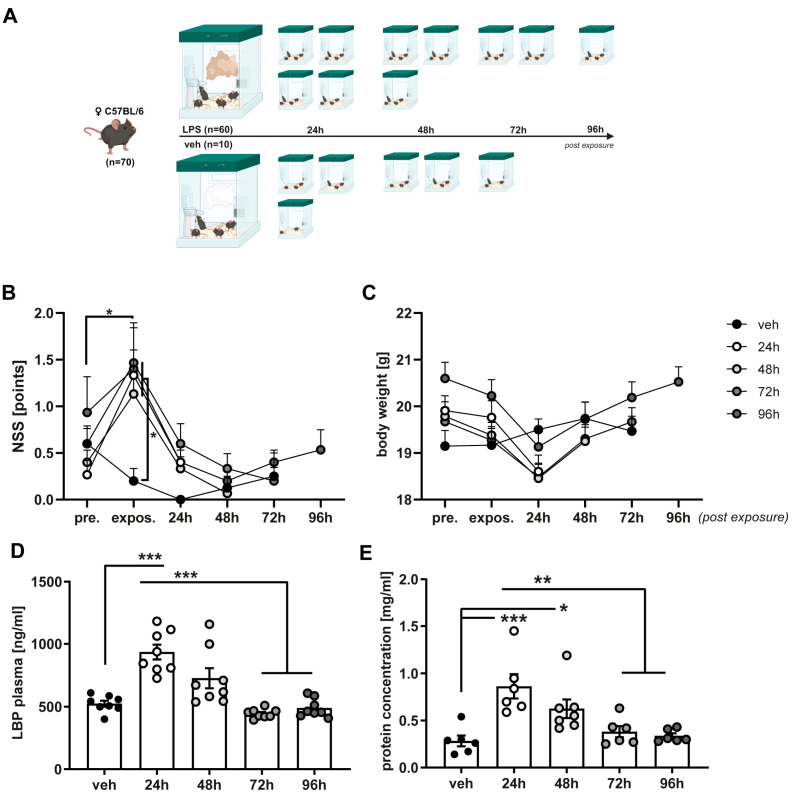
Single LPS exposure exerts short-term effects on neuromotor performance and translocates the inflammatory stimulus to the bloodstream. (**A**) Schematic description of the study design. Female C57BL/6 mice were exposed to nebulized lipopolysaccharide (LPS) or vehicle solution (veh) and randomized in groups with 24 h, 48 h, 72 h or 96 h survival. (Created in BioRender. Ritter, K. (2024) https://biorender.com/l20m471 (accessed on 15 September 2024)). (**B**) Neuromotor deficit was assessed repetitively by Neurological Severity Score (NSS). All mice subjected to nebulized LPS showed increased scores indicating greater impairment the same day after exposure (expos.) in comparison to their pre-exposure (pre.) performance and vehicle, with this effect subsiding from 24 d after exposure. (**C**): Body weight was registered daily during the experiment. Groups subjected to LPS showed a reduction in body weight 24 h after exposure, yet this observation did not obtain a statistical level of significance. (**D**): Plasma concentrations of lipopolysaccharide-binding protein (LBP) were determined by ELISA in all groups (*n =* 8 each). LBP levels were significantly increased 24 h and 48 h after exposure in comparison to vehicle as well as 72 h and 96 h. (**E**): Protein concentration in bronchoalveolar lavage (BAL) was quantified by Lowry assay in all groups (*n =* 7 each). Mice 24 h and 48 h after exposure showed significantly increased protein levels in BAL compared to the vehicle as well as 72 h and 96 h after exposure. * *p* < 0.05, ** *p* < 0.01, *** *p* < 0.001. Values of all data represent mean ± SEM; *p* values were calculated by One-way- (**D**,**E**) or Two-way-ANOVA (**B**,**C**) followed by Holm–Sidak’s multiple comparison test.

**Figure 3 ijms-25-10117-f003:**
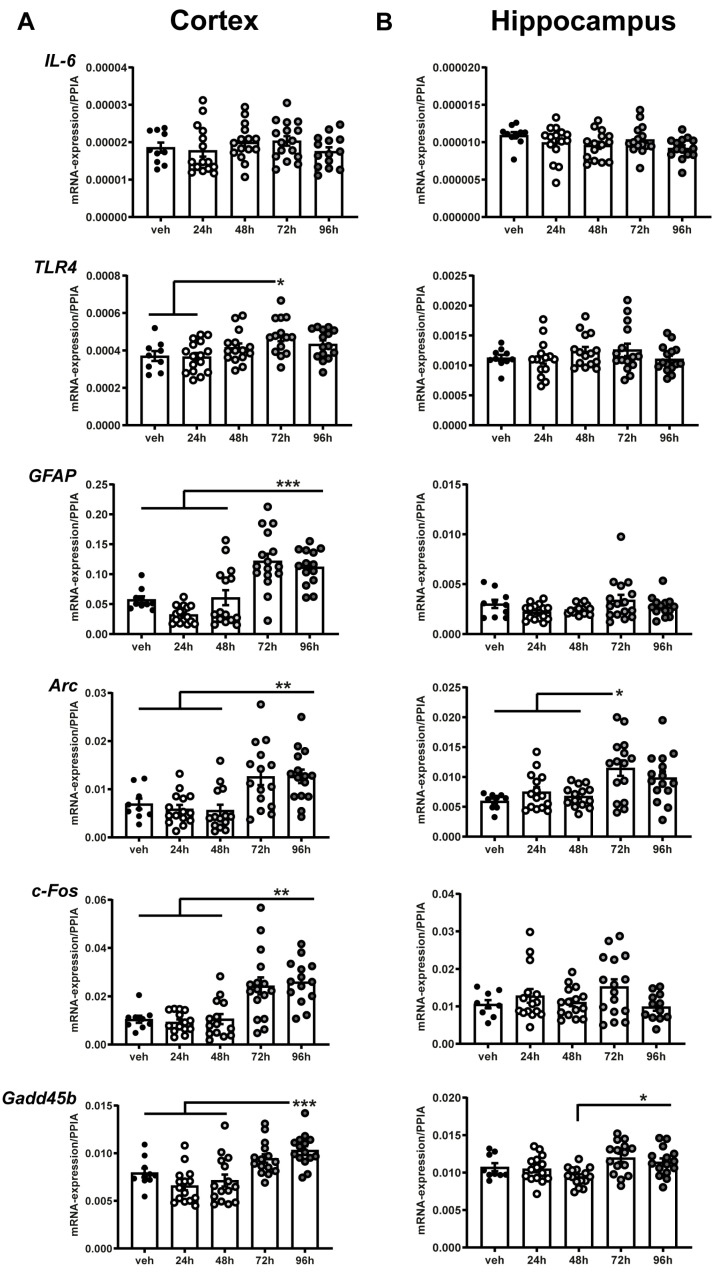
Single exposure with nebulized LPS leads to delayed neuroinflammatory gene expression in the cerebral cortex and hippocampus. Gene expression analyses for several markers of inflammation and neuroinflammation in tissue samples of cerebral cortex (**A**) and hippocampus (**B**) were performed by qPCR in all study groups and normalized to *PPIA*. While mRNA expression of *IL-6* was not induced by LPS exposure in both regions, expression of *GFAP* and *c-Fos* was increased 72 h and 96 h after LPS exposure in comparison to vehicle as well as 24 h and 48 h after nebulization in the cortical brain, yet unaffected in the hippocampus. Expression of *Gadd45b* was strongly increased 72 h and 96 h after LPS exposure in the cortical brain and also elevated in the hippocampus compared to vehicle and the earlier examination time points. Expression of the *Arc* gene was significantly increased in the cortex 72 h and 96 h and in the hippocampus 72 h after exposure in comparison to vehicle as well as 24 h and 48 h after nebulization. * *p* < 0.05, ** *p* < 0.01, *** *p* < 0.001. Values of all data represent mean ± SEM; *p* values were calculated by One-way-ANOVA followed by Holm–Sidak’s multiple comparison test.

**Figure 4 ijms-25-10117-f004:**
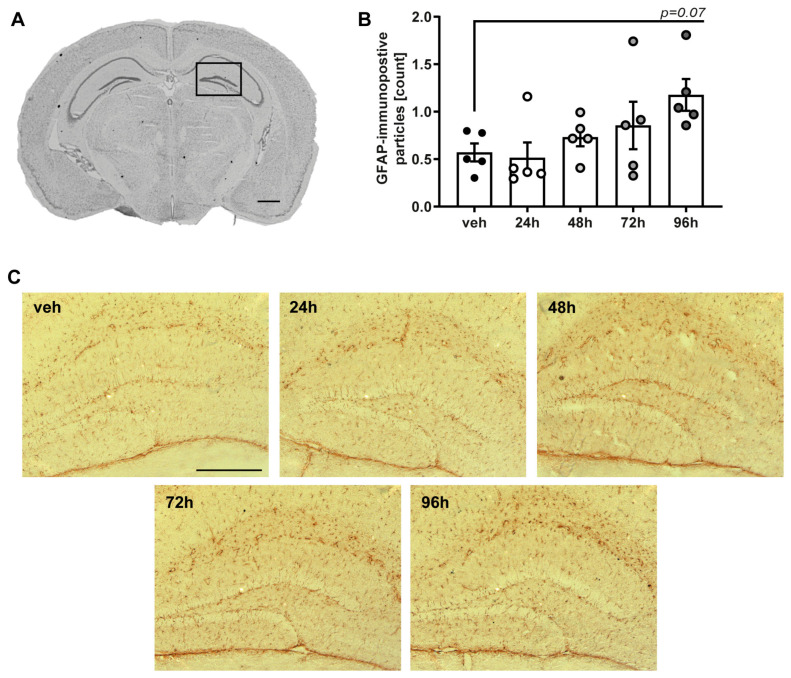
Single LPS-exposure induces astrocytic response in the dentate gyrus. (**A**): Exemplary demonstration of the analyzed hippocampal brain region (bregma −2.00 mm, scale bar 1 mm). (**B**): Count of GFAP immunopositive particles tended to be increased 96 h after LPS nebulization compared to vehicle. (**C**): Representative images of GFAP-immunostaining in the dentate gyrus of all study groups. Scale bar: 200 μm. Values of all data represent mean ± SEM; *p* values were calculated by One-way-ANOVA followed by Holm–Sidak’s multiple comparison test.

**Table 1 ijms-25-10117-t001:** Gene name, amplicon size, and oligonucleotide sequences (brain, lung).

Gene Name, (Amplicon Size, bp)	Oligonucleotide Sequences 5′–3′ (fw: Forward, rev: Reverse)	Gene Bank Number
*Arc* (192)	fw-CTCAACTTCCGGGGATGCAG rev-CTGGTATGAATCACTGGGGGC	NM_001276684
*Fos* (165)	fw-CGGGTTTCAACGCCGACTA Rev-TGGCACTAGAGACGGACAGAT	NM_010234
*Gadd45b* (113)	fw-CCTCCTGGTCACGAACTGTC rev-TGGGTCTCAGCGTTCCTCTA	NM_008655
*Gfap* (120)	fw-CGGAGACGCATCACCTCTG rev-TGGAGGAGTCATTCGAGACAA	NM_001131020
*IL-6* (471)	fw-CATAAAATAGTCCTTCCTACCCCAATTTCC-FL rev-TATGCTTAGGCATAACGCACTAG	NM_031168
*Ppia* (146)	fw-GCGTCTSCTTCGAGCTGTT rev-RAAGTCACCCTGGCA	NM_008907
*TLR4* (134)	fw-GCTTTCACCTCTGCCTTCAC rev-CCAACGGCTCTGAATAAAGTG	NM_021297.3
*TNFalpha* (212)	fw-TCTCATCAGTTCTATGGCCC rev-GGGAGTAGACAAGGTACAAC	NM_013693

## Data Availability

The datasets generated and analyzed during the current study are included in this published article or available from the corresponding author upon reasonable request.

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
