# Peer review of "Nebulized Lipopolysaccharide Causes Delayed Cortical Neuroinflammation in a Murine Model of Acute Lung Injury"

_ijms, 2024, doi:10.3390/ijms251810117_

Round 1

Reviewer 1 Report

Comments and Suggestions for Authors

In their manuscript Ritter et al. mimicked a mouse model of LPS-dependent lung injury without the need of a nasal or intratracheal application of LPS but using a nebulized LPS preparation with a 30 min full-body incubation of the mice. The authors observed an increase in pro-inflammatory cytokine expression mainly after 24 h following LPS exposure in the lung. An induction of pro-inflammatory markers in the brain did not change (e.g. IL-6) or increased after alonger period of time such as e.g. 72 h (GFAP, c-Fos) at least in the cortex.

Although the used approach is very gentle, the amount of LPS taken up by each of the female mice is not validated. The author determined the amount of LBP in the plasma as a readout for the systemic effect of LPS, which might also make an impact on the brain.

The study is of general interest, because mouse models with a gentle impact on the mouse comfort, should be preferred. However, in my opinion the relationship of the full-body exposure of nebulized LPS to intranasal, intratracheal, and intraperitoneal application, or if you are only interest in the brain intracerebraventricular is not clear. One experiment estimating the amount of LPS which is inhaled by the animals should be included and discussed comparing the other, already establsihed more invasive LPS application methods.

Author Response

Comment 1: In their manuscript Ritter et al. mimicked a mouse model of LPS-dependent lung injury without the need of a nasal or intratracheal application of LPS but using a nebulized LPS preparation with a 30 min full-body incubation of the mice. The authors observed an increase in pro-inflammatory cytokine expression mainly after 24 h following LPS exposure in the lung. An induction of pro-inflammatory markers in the brain did not change (e.g., IL-6) or increased after alonger period of time such as e.g. 72 h (GFAP, c-Fos) at least in the cortex.

Although the used approach is very gentle, the amount of LPS taken up by each of the female mice is not validated. The author determined the amount of LBP in the plasma as a readout for the systemic effect of LPS, which might also make an impact on the brain.

The study is of general interest, because mouse models with a gentle impact on the mouse comfort, should be preferred. However, in my opinion the relationship of the full-body exposure of nebulized LPS to intranasal, intratracheal, and intraperitoneal application, or if you are only interest in the brain intracerebraventricular is not clear. One experiment estimating the amount of LPS which is inhaled by the animals should be included and discussed comparing the other, already establsihed more invasive LPS application methods.

Response 1: We thank the reviewer for this very important comment. We fully agree, that the missing validation of the exact amount of LPS taken up by each mice is a major limitation of this study compared to more invasive models of LPS administration. The inflammatory gene expression in the lung and the increase of LBP plasma levels serve as an indirect evidence of a sufficient stimulus, yet the dose of LPS, which was necessary to induce this response, remains unknown. We ask to consider, that the exact amount of applied LPS is only known in models of injection or direct intratracheal instillation under anaesthesia – the administrations intranasal or via oropharyngeal aspiration also bear the risk of an uncertain amount of LPS not reaching its destination. To determine the exact lung deposition of LPS (e.g. by ELISA) an analysis of the complete organ is necessary, and the samples in this study were used for BAL, qRT-PCR and histological analysis. The IJMS deadline for re-submission lies within ten days after receipt of the reviews. We kindly ask for understanding that a separate experiment is not practicable within this time period, especially given the circumstance, that we would need to apply for permission to use more animals in this project. Nevertheless, the amount of LPS reaching the lung can be estimated on a theoretical level. The full-body-exposure was performed in a sealed cage with a volume of 8.5 litres and a total of 15 mg LPS, which was fully nebulized during the first 15 minutes, while mice rested the second 15 min of the 30min period in the aerosol. This leads to concentrations from 0.11 μg LPS/ml air at the beginning to a maximum of 1.76μg LPS/ml air at the moment of full nebulization. Calculating a respiratory rate of 200/min with a 150 μl tidal volume in adult mice, the breathing volume per minute results in 30 ml [60] leading to a maximum of 52 μg LPS per minute circulating through the airway. Given the fact, that according to the manufacturers instruction the nebuliser produces an aerosol with 90% particle size above 15 μm and larger particles tend to faster deposition, the real amount might be lower. The more invasive models of intratracheal or intranasal administration use doses from 50 μg to 75 μg [37,38], for intraperitoneal injection the amount is about 2 μg/g body weight (approximately 40-50 μg LPS in total). Regarding the exposure period of 30 min in our experiment, the amount of LPS taken up by the individual animal might be higher than in the invasive models. Nevertheless, none if the mice in our experiment met termination criteria and we observed a significant decrease of the inflammatory response after 48 h. The chosen full-body-exposure does not require anaesthesia nor does have the side effect of flushing the endogenous barrier of the lung endothelium by aspiration and therefore remains preferable for the authors, yet dose-dependent analyses should be performed in future project.

We added the following sections to the manuscript:

“Nebulization was performed by a medical nebulizer (IH50, Beurer, Ulm, Germany) placed in the temporarily sealed cage with a volume of 8.5 liters for 30 min as full-body-exposure. Process of nebulization required 15 min, leading to concentrations from 0.11 μg LPS/ml air at the beginning to a maximum of 1.76μg LPS/ml air at full nebulization, and mice rested in the aerosol for 15 min hereafter. Considering the standard respiratory rate of adult mice this leads to a theoretical maximum of 52 μg LPS/min circulating through the airway.” (Material and methods, lines 287-294)

We further added the unknown amount of applied LPS as a major limitation in the discussion:

“A major limitation of the study is that the amount of LPS taken up by each animal re-mains unknown. The concentration of the aerosol is estimated, yet the amount of circulating LPS that depletes in the lung is not determined. Considering the calculated concentrations and duration of exposure, the LPS dose in this study might potentially exceed that of more invasive models utilizing intratracheal instillation or i.p. injection [37, 38]. However, none of the animals in this study met termination criteria and we observed a significant decrease of the inflammatory response in the lung after 48 h. (Discussion, lines 203-209)

Reviewer 2 Report

Comments and Suggestions for Authors

Comments:

   The manuscript describes " Nebulized Lipopolysaccharide Causes Delayed Cortical Neuroinflammation in a Murine Model of Acute Lung Injury” Lung damage from respiratory infections is a leading cause of hospitalization and death and a leading cause of sepsis. This experiment utilized C57BL/6 mice that received a single systemic exposure to aerosolized lipopolysaccharide. The neuromotor injury was scored repeatedly, and brain, blood, and lung samples were analyzed at various survival points at 24, 48, 72, and 96 hours after exposure. showed increased expression of TNFα and IL-1β in the lungs 24 and 48 hours after LPS exposure, along with increased protein content in bronchoalveolar lavage fluid and interstitial pulmonary edema. In the cerebral cortex, inflammation and activity-related markers increased at 72 and/or 96 h after LPS exposure. A single exposure to aerosolized lipopolysaccharide causes an early peak of inflammation in the lungs, translocates the irritant into the bloodstream, and induces a delayed neuroinflammatory response in the cerebral cortex., but several points need clarification.

Comment:

1. In the figure1, image of HE staining of lung tissue samples. The author should perform statistical analysis

2. A single LPS exposure produces short-term effects on neuromotor performance and translocates inflammatory stimuli into the bloodstream. Why are NSS and BW not time-dependent? The author should discuss this carefully.

3. Authors of GFAP immunostaining should carefully analyze statistics.

4. Is there any dependence between the dose and time of aerosolized lipopolysaccharide and Neuroinflammation?

5. Why were these 8 inflammation targets chosen? qRT-PCR was performed on cortical and hippocampal brain tissue samples. The author should discuss this carefully.

Comments on the Quality of English Language

Minor editing of English language required.

Author Response

Comment 1: In the figure1, image of HE staining of lung tissue samples. The author should perform statistical analysis
Response 1: HE staining of lung tissue samples was analysed by histological scoring and quantification of the tissue free area (Figure 1 C, D). All criteria were analysed statistically (Methods “4.5 Statistical analyses”) with levels of significance deployed in the graph and the detailed results in the result section (lines 92-97).

Comment 2: A single LPS exposure produces short-term effects on neuromotor performance and translocates inflammatory stimuli into the bloodstream. Why are NSS and BW not time-dependent? The author should discuss this carefully.
Response 2: We thank the reviewer for this mindful objection. As declared in the discussion, the chosen assessment of the neuromotor impairment in this study is insufficient to detect dysfunctions caused by a diffuse cortical neuroinflammation and therefore must be considered as a major limitation. The impaired performance in the Neurological Severity Score at the day of exposure is most likely caused by a reduction of general comfort in the LPS-exposed mice. As our previous presentation was unclear, we addressed this topic more precisely and revised the following section in the discussion:

“Neuromotor impairment was analysed by NSS, a tool frequently used in models of experimental TBI [32-34]. All mice subjected to nebulized LPS showed significantly in-creased scores compared to vehicle at the day of exposure, yet this effect faded rapidly within 24 h. While NSS serves well in discriminating the degree of motoric dysfunction after a localized cerebral injury, its capacity of capturing higher cognitive functions is strictly limited. Most studies about LPS-induced neuroinflammation or SAE include assessment of spatial learning, long-term memory and emotional learning [22, 23, 27]. As the observed impact on the NSS appeared at the day of exposure and far before the up-regulation of neuroinflammatory gene expression, it is most likely confounded by a temporary debilitated general condition. Therefore, not addressing more elaborated functions in neurobehavioural testing is considered as a major limitation of this study.” (lines 214-224)

Comment 3: Authors of GFAP immunostaining should carefully analyze statistics.
Response 3: The results of GFAP immunostaining were statistically analysed as described in Methods “4.5 Statistical analyses”.  According to their distribution, data were analysed by One-way-ANOVA followed by Holm-Sidak's multiple comparison test as mentioned in figure legends of figure 4. The statistical result shows a trend (p=0.065), yet is not significant. In the result section, it is already described as “trend towards statistical significance”. (lines 177-180). To clarify this statement in the figure, we changed the figure legend to:

“Count of GFAP immunopositive particles tended to be increased 96 h after LPS nebulization in comparison to vehicle.”

Comment 4: Is there any dependence between the dose and time of aerosolized lipopolysaccharide and Neuroinflammation?
Response 4: We appreciate the remark about dose dependence of dose and time in this project. In our study, only a single dose of LPS with a consistent application method was used. Regarding the presented findings, it is quite understandable to question if different doses could lead to earlier or later responses. Unfortunately, our study cannot provide such information.

We thank the reviewer for pointing out this limitation, and added it to the discussion as it follows:

“This study worked with a consistent dose of LPS and therefore cannot provide a state-ment about the onset of neuroinflammation in higher or lower doses of LPS.” (lines 260-261)

Comment 5: Why were these 8 inflammation targets chosen? qRT-PCR was performed on cortical and hippocampal brain tissue samples. The author should discuss this carefully.
Response 5: We are grateful for this comment and agree that the selection of markers and their previously described expression profiles in the specific brain regions require more detailed information. We now write in the results section:

“qRT-PCR was performed for several (neuro-)inflammatory targets in brain tissue samples of cortex (Figure 3A) and hippocampus (Figure 3B) from all analysed survival points. IL-6 was chosen as acute phase marker. Toll-like-receptor 4 (TLR4) is the main receptor for LPS, which mediates pro-inflammatory signaling in various neurological disorders [28] and glial fibrillary acidic protein (GFAP) is a common marker for astro-cyte activation [29]. Activity-regulated cytoskeletal gene (Arc), c-Fos and growth arrest and DNA damage inducible beta (Gadd45b) are immediate early genes (IEGs) and are considered as markers upregulated in response to physiological and environmental stressors. While Arc is predominantly expressed by neurons and critically involved in synaptic plasticity [30], c-Fos and Gadd45b are expressed by neurons and glia, serving both as markers for neuronal and glial activation [31, 32]. (lines 144-145)

Moreover, we have revised the discussion of the results of the gene expression analysis and agree that this part required careful revision. We now write:

“Our analyses revealed increased expression of GFAP, Arc, Gadd45b and c-Fos in the cerebral cortex at 72 h and 96 h after exposure. The increased expression of GFAP indicates the activation of astrocytes [29], which was induced by i.p. administration of LPS and connected to NFκB downstream signaling in previous works, while in vitro settings revealed CD14 as a dominant co-factor in TRL4 mediated astrocytic stimulation [42-44]. However, the delayed up-regulation of GFAP mRNA-expression was only ob-served in the cortex, but not in the hippocampus. A similar regulation was detected for the mRNA-expression of the IEGs c-Fos and Gadd45b, which serve as indicators for cellular responses to environmental stressors [31, 32]. c-Fos was shown to be upregu-lated after LPS stimulation in primary glia cultures and astrocytes in vitro [45, 46], whereas upregulation of Gadd45b was associated with anti-apoptotic processes in as-trocytes [47]. Together, the mRNA-expression profile of GFAP, c-Fos and Gadd45b indicate a delayed glial activation in the cerebral cortex in response to nebulized LPS ex-posure.” (lines 225-237)

Reviewer 3 Report

Comments and Suggestions for Authors

Comments and suggestions:

1. Overall structure and clarity:
The paper is generally well-structured, following a standard scientific format with clear sections. However, the introduction could be more focused and concise. It covers a broad range of topics without always making clear connections to the study's specific aims.

2. Abstract:
The abstract is informative but could be more concise. It currently exceeds 250 words, which is longer than many journals allow. Consider trimming some details to focus on the most crucial information.

3. Methods:
The methods section is detailed, which is good for reproducibility. However, some subsections (e.g., histological analyses) are quite long and could potentially be streamlined.

4. Results:
The results are presented clearly, with appropriate use of statistics. However, some figures (e.g., Figure 1) are quite dense with information and may be difficult for readers to interpret quickly. Consider simplifying or splitting complex figures.

5. Discussion:
The discussion does a good job of interpreting the results in the context of existing literature. However, it could be strengthened by more clearly stating the novel contributions of this study and addressing potential limitations.

6. Language and style:
While generally well-written, there are occasional awkward phrasings or grammatical issues. For example, in the abstract: "Sepsis-associated encephalopathy and delirium are frequent complications in patients with severe lung injury and acute respiratory distress syndrome, yet the pathogenetic mechanisms are not fully understood." This sentence is long and could be split for clarity.

7. References:
The reference list appears comprehensive and up-to-date. However, ensure all citations in the text match the reference list.

8. Figures and Tables:
The figures are informative, but some (like Figure 1) are very dense with information. Consider ways to simplify or split complex figures for easier interpretation.

9. Statistical analysis:
The statistical methods seem appropriate, but more details on power analysis or sample size determination would be beneficial.

10. Conclusion:
The conclusion is somewhat brief. It could be expanded to more clearly state the implications of the findings and suggest future research directions.

Overall, this is a solid scientific manuscript with interesting findings. With some refinement in presentation and clarity, it could make a valuable contribution to the field.

Author Response

Comment 1: Overall structure and clarity: The paper is generally well-structured, following a standard scientific format with clear sections. However, the introduction could be more focused and concise. It covers a broad range of topics without always making clear connections to the study's specific aims.
Response 1: Overall, we thank the reviewer for his constructive comments about the manuscripts structure and writing (1.-6.). We performed minor corrections to the introduction in order to strengthen it, yet kindly ask the reviewer to consider the aspect that this manuscript delves into the mechanisms of two diseases (ALI and SAE). We hope for understanding, that we aimed at addressing both pathologies properly.

Comment 2: Abstract: The abstract is informative but could be more concise. It currently exceeds 250 words, which is longer than many journals allow. Consider trimming some details to focus on the most crucial information.
Response 2: We thank the reviewer for this observation. The abstract is now reduced to 183 words and focuses on the main finding of the chronological divergence of lung and brain inflammatory response.

Comment 3: Methods: The methods section is detailed, which is good for reproducibility. However, some subsections (e.g., histological analyses) are quite long and could potentially be streamlined.
Response 3: We agree with the reviewer, that specific method sections are long and very detailed as the authors aimed to provide full insight in the used methods. As proposed by the reviewer, the segment “4.2 Histological analyses” has been trimmed as it follows:

“3,3′-Diaminobenzidine (DAB)-based immunostaining was used to detect GFAP-immunopositive particles in the right hippocampus as previously described [62, 63]. Cryosections were fixed in 4% paraformaldehyde in phosphate-buffered saline (PBS), rinsed in PBS and incubated in 3.0% H2O2. Sections were incubated with blocking solution (5% normal goat serum, 1% bovine serum, and 0.1% Triton-X100 in PBS) for 1 h at room temperature (RT). Primary antibody (rabbit anti-GFAP Z033401-Z, DAKO, Santa Clara, CA, USA, 1:500) was applied in blocking solution and incubated at 4 °C overnight. The next day, sections were washed in PBS and incubated with secondary antibody (goat anti rabbit (H + L) biotinylated #BA-1000-1,5, 1:50, Vector Laboratories, Newark, CA, USA) for 1.5 h at RT. Samples were then incubated with avidin-biotin complex (1 drop avidin, 1 drop biotin, 2.5 ml PBST; Vectastain ABC Kit, Vector Laboratories) for 2 h at RT, washed and incubated with 3,3′-diaminobenzidine (1 drop for 1 ml PBS). Sections were washed and air dried, fixed in Xylol (Pancreac AppliChem, Darmstadt, Germany) and mounted in Entellan (VWR, Darmstadt, Germany).” (lines 319-332)

Comment 4: Results: The results are presented clearly, with appropriate use of statistics. However, some figures (e.g., Figure 1) are quite dense with information and may be difficult for readers to interpret quickly. Consider simplifying or splitting complex figures.
Response 4: We agree with the reviewer, that some of the figures provide a dense amount of information. The arrangement of the figures follows their specific subject (e.g., Figure 1 demonstrates the organic response of the lung, Figure 2 shows clinical data, Figures 3 and 4 depict gene expression and histological findings in the brain). All figures fulfil IJMS criteria. We fear, that splitting up figures might interrupt the process of reading and kindly hope for understanding that we chose to keep the original design.

Comment 5: Discussion: The discussion does a good job of interpreting the results in the context of existing literature. However, it could be strengthened by more clearly stating the novel contributions of this study and addressing potential limitations.
Response 5: We agree with the reviewer, that the discussion addresses several topics of interest. We revised the discussion in reference to point 6. and restructured particular text segments in order to provide more clarity. The novel contributions of this study lie in the induction of neuroinflammation via addressing the lung as primary organ and the chronological delay between pulmonary and cerebral response. Those aspects are pointed out in the discussion and conclusion:

“Taken together, the single full-body-exposure to nebulized LPS provides a sufficient inflammatory lung injury and therefore represents a less invasive alternate model to the predominantly used intratracheal instillation. Future studies are needed to analyse dose dependent effects on the inflammatory response and to quantify the exact amount of LPS supply.” (Discussion, lines 209-213)

“Taken together, we detected a delayed upregulation of several markers of neuronal and glial activation and inflammation in the cerebral cortex, but also in the hippocampal area following the induction of a pulmonary inflammation by nebulized LPS.” (Discussion, lines 252-255)

“The specificity of this study lies in the way of application of the pro-inflammatory stimulus and the time-lapse between the inflammatory responses of the particular organs, as neuroinflammation-associated gene expression in the brain started to appear 72 h after the exposure.” (Conclusion, lines 396-399)

The limitations of the study were pointed out more concrete as it follows:

“A major limitation of the study is that the amount of LPS taken up by each animal re-mains unknown. The concentration of the aerosol is estimated, yet the amount of circulating LPS that depletes in the lung is not determined.” (Discussion, lines 203-205)

“Therefore, not addressing more elaborated functions in neurobehavioural testing is considered as a major limitation of this study.” (Discussion, lines 223-224)

“This study worked with a consistent dose of LPS and therefore cannot provide a statement about the onset of neuroinflammation in higher or lower doses of LPS.” (Discussion, lines 260-261)

Comment 6: Language and style: While generally well-written, there are occasional awkward phrasings or grammatical issues. For example, in the abstract: "Sepsis-associated encephalopathy and delirium are frequent complications in patients with severe lung injury and acute respiratory distress syndrome, yet the pathogenetic mechanisms are not fully understood." This sentence is long and could be split for clarity.
Response 6: We appreciate the remark about structure and wording. The quoted sentence was altered to “Sepsis-associated encephalopathy and delirium are frequent complications in patients with severe lung injury, yet the pathogenetic mechanisms remain unclear.” Several other sentences, which appear long or twisted have been revised in all sections of the manuscript.

Comment 7: References: The reference list appears comprehensive and up-to-date. However, ensure all citations in the text match the reference list.
Response 7: The references were selected carefully by their significance to the manuscript’s topic and authors were dedicated to include the most recent publications. All citations in the text and their matching to the reference list were checked properly before re-submission.

Comment 8: Figures and Tables: The figures are informative, but some (like Figure 1) are very dense with information. Consider ways to simplify or split complex figures for easier interpretation.
Response 8: As the structure of the figures is already addressed before, we kindly ask to refer to the answer of “4. Results”.

Comment 9: Statistical analysis: The statistical methods seem appropriate, but more details on power analysis or sample size determination would be beneficial.
Response 9: We fully appreciate the reviewer’s request to describe sample size determination and power analyses in detail. To address this topic properly, we added the following section:

“As this is an exploratory study, previous data, which could be used to calculate group sizes, are rare. The required sample size (n = 15 animals in LPS groups, n = 10 in the vehicle group) was determined based on the assumption that a change of 20 % in the analysed criteria is relevant. The probability of type I error was set to a = 0.05, and the standard statistical power was set to 1-b = 0.8 (80%) resulting in b = 0.2 (probability of type II error). A larger amount of animals was used in LPS groups due to the natural range of inflammatory response, which had not to be expected in the vehicle group.” (Material and Methods, lines 376-382)

Comment 10: Conclusion: The conclusion is somewhat brief. It could be expanded to more clearly state the implications of the findings and suggest future research directions.
Response 10: We thank the reviewer for focusing the authors attention on the conclusion and agree fully about the lack of extent. We revised the conclusion section as it follows:

“Taken together, this study confirmed a single full-body exposure to nebulized LPS as an effective inflammatory stimulus to the lung, able to cross the barriers between lung, brain and circulating bloodstream. The specificity of this study lies in the way of application of the pro-inflammatory stimulus and the time-lapse between the inflammatory responses of the particular organs, as neuroinflammation-associated gene ex-pression in the brain started to appear 72 h after the exposure. These findings might en-lighten the pathologic background of neurocognitive disorders in acute lung injury. Further work is required to explore the impact on neurocognitive function and identify potential therapeutic approaches.” (Conclusion, lines 394-402)

Round 2

Reviewer 1 Report

Comments and Suggestions for Authors

All of my concerns have been successfully addresses by the authors.

Reviewer 2 Report

Comments and Suggestions for Authors

Accepted

Comments on the Quality of English Language

Minor editing of English language required.